# Parents of children with hearing loss: Impact and exposure of COVID-19 on mental health

**Ivette Cejas** *, **Chrisanda Sanchez, Meredith Holcomb, Jennifer Coto**

Department of Otolaryngology, University of Miami Miller School of Medicine, Miami, FL, United States of America

* icejas@med.miami.edu

**Data Availability Statement:** All relevant data are within the paper and its Supporting Information files.

**Funding:** Funded by NIH R01DC012115-08S1 (IC) & University of Miami Sharing Auditory Miracles

## Abstract

The aim of this study was to evaluate the impact and exposure of COVID-19 on parent mental health (e.g., depression, anxiety, and post-traumatic stress disorder (PTSD), for parents of children with hearing loss. The survey was distributed via an electronic survey to families subscribed to a pediatric program listserv as part of a university medical center. Fifty-five percent of parents reported elevated symptoms of anxiety, while 16% scored in the clinically significant range for depression. In addition, 20% of parents reported elevated symptoms of PTSD. Liner regressions found that impact of COVID-19 predicted anxiety symptoms, while both impact and exposure predicted depression and PTSD symptoms. In addition, both impact and exposure predicted COVID related parental distress. Exposure and impact of COVID-19 has had negative consequences on parents of children with hearing loss. Although exposure influenced parental mental health, impact uniquely affected depression and PTSD. Results highlight the need for mental health screening, as well implementation of psychological interventions using telehealth or in-person consultations. Future work should focus on post-pandemic challenges, including long-term psychological functioning due to the established relationship between parental mental health and pediatric outcomes.

## Introduction

The coronavirus disease 2019 (COVID-19) has drastically affected the world, with studies reporting the negative impact on mental health in both children and parents [1,2]. Importantly, these mental health consequences are not evenly distributed across populations as some are at higher risk than others. Children with hearing loss and their families are considered an at-risk population for increased rates of depression and anxiety [3] due to their communication, social and emotional delays [4]. Likewise, parents who are depressed are three times more likely to have children with mental health disorders, as well associated cardiovascular and neuromuscular conditions [5]. However, little is known about how parental mental health during a traumatic event can affect child mental health and outcomes in families of children with hearing loss. Families have experienced substantial hardships throughout the pandemic including illness, job loss, difficulty with obtaining child-care, and caregiver burden, all of which can add to parenting stress. This is particularly important for the at-risk population of

(SAM) Fund. The funders had no role in study design, data collection and analysis, decision to publish, or preparation of the manuscript.

**Competing interests:** The authors have declared that no competing interests exist.

children with hearing loss as parenting stress can negatively impact pediatric hearing loss outcomes [6]. In general, parents of children with hearing loss have reported higher levels of parenting stress and less parental sensitivity than parents of children with normal hearing, with those parents reporting more stress and less sensitivity having children with more substantial speech and language delays [7,8]. If left unaddressed, parenting stress has the potential to cause greater tension in the home and can result in potential child maltreatment [9]. Specifically, mental health symptoms have also been shown to affect treatment adherence and parent involvement, both of which affect overall child outcomes [10–12]. Thus, it is imperative that we begin to evaluate the mental health of parents caring for children with hearing loss, particularly during this heighted period of stress due to the documented increased rates of mental health conditions worldwide [13].

This study evaluated the impact (wellbeing, emotional adjustment, sleeping and ability to care for children) and exposure (school closures, changes in housing, changes in employment) of COVID-19 on the overall mental health of parents of children with hearing loss, as defined by symptoms of depression, anxiety, and Post-Traumatic Stress Disorder (PTSD). This study will not only help disentangle and better understand the psychological sequelae of the pandemic in a high-risk population but will also identify whether additional supports are needed for parents of children with hearing loss. We hypothesized that there would be higher symptoms of depression, anxiety and PTSD in this sample compared to the general population. Additionally, we hypothesized that increased impact and exposure of COVID-19 would be associated with mental health symptoms. Lastly, we predicted that impact of COVID-19 would be associated with parental distress.

## Materials and methods

Participants were 103 parents of children with hearing loss who were established patients and seeking treatment/intervention for their hearing loss from a comprehensive pediatric otology/audiology practice. All families were recruited from the program's pediatric listserv. The clinic serves mostly a Hispanic community without private insurance and is located in a large metropolitan hospital in the southeastern United States. This study was approved by the University of Miami Institutional Review Board. Primary caregivers, per our pediatric listserv, received an email containing a Qualtrics (online survey platform) link in English and Spanish that provided information regarding the study, as well as consent to participate. Consent was obtained and documented during the electronic survey. The first page of the online questionnaire was the informed consent. Participants indicated consent by clicking "yes" at the bottom of the consent form. If participant selected "no" the survey was immediately ended. This consent process was approved by the University of Miami Institutional Review Board. The survey was open for a period of one month (June 9th, 2020- July 9th, 2020) and included a self-reported demographics section [e.g., age, gender, race/ethnicity, child hearing device type] as well as several measures assessing COVID-19 Exposure and Impact, parent mental health, and parenting stress. Participants were permitted to skip questions they did not wish to answer. Participants received a $25 electronic gift card after completion of the survey.

### Sample characteristics

The sample consisted of 103 parents of children with hearing loss. The majority of parents were White (80.8%), Hispanic (67.1%), mothers (86.4%) with a mean age of 38.39 ($SD$ = 8.70) who were married (62.1%). The survey was completed by 82.5% of parents in English and the primary language children spoke at home was predominantly oral/spoken language (89.3%). Over half of parents (65%) were working on-site, from home or a combination, while 10.7%

had been laid off or furloughed. Children had a mean age of 9.72 (*SD* = 4.16) and most wore cochlear implants (52.4%) or hearing aids (37.9%). All demographic and audiological information was obtained via parent report via the online survey. See Table 1 for further descriptive information.

## Measures

**COVID-19 exposure and family impact survey.** The COVID-19 Exposure and Family Impact Survey (CEFIS) is a validated measure that assesses exposure to potentially traumatic aspects of COVID-19 and the impact of the pandemic on the family unit [14; see Fig 1]. Part 1 (exposure) consists of 25 items with yes or no responses. Exposure to COVID-19 and related events was defined as being exposed to someone in the family that was diagnosed with COVID-19, school closures, changes in housing, changes in employment, or inability to visit or care for family members (e.g., "Our family income decreased;" "Someone in the family was exposed to someone with COVID-19"). Part 2 (impact) consists of 12 items, 10 of which use a 4-point Likert scale (1 = Made it a lot better to 4 = Made it a lot worse) and comprise the Impact scale, and two that use a 10-point distress scale (1 = No distress to 10 = Extreme distress) and individually comprise a parent distress scale and child distress scale. Impact of COVID-19 reflects an individuals' overall wellbeing, emotional adjustment, sleeping, and ability to care for their children (e.g., "In general, how has the COVID-19 pandemic affected each of the following: parenting; . . .how family members get along with each other"). These single item distress scales have demonstrated reliability and criterion validity. Part 3 is an open-ended question inquiring about the family's experience and effects of COVID (e.g., "Please tell us about other effects of COVID-19 on your child/ren and your family, both negative and/or positive"); this qualitative question was not used in the current study. Higher scores on the exposure, impact, and distress scales denote more negative exposure, impact, and distress. The Exposure scale (α = .60), Impact scale, (α = .91) and distress questions were utilized in this study.

**Generalized anxiety disorder.** The Generalized Anxiety Disorder [GAD-7; 15] is a validated and widely used 7-item self-report questionnaire of anxiety symptomatology in the past 14 days. Participants indicate the frequency of each item during the past two weeks and include feelings of nervousness, worry, irritability, or trouble relaxing. Items are rated on a 4-point Likert scale, ranging from 0 = Not at all to 3 = Nearly every day. The sum of the items indicates the overall level of anxiety symptoms and are categorized into mild (0–4), moderate (5–9), moderately severe (10–14), and severe (15–21). Scores beginning at the moderately severe range are considered clinically significant and warrant further evaluation for anxiety (α = .92). Evidence supports validity and reliability of the GAD-7 with various populations and in primary care settings [15].

**Patient health questionnaire.** The Patient Health Questionnaire [PHQ-8; 16] is a validated 8-item self-report questionnaire used to assess depressive symptoms. Participants indicate how bothered they have been over the past two weeks for problems such as, feeling down, tired, or feeling bad about themselves. Items are rated on a 4-point Likert scale (0 = Not at all to 3 = Nearly every day) with higher scores indicating greater frequency of depressive symptoms. The sum of the items indicates the overall level of depressive symptoms indicating mild (5–9), moderate (10–14), moderately severe (15–19), and severe depressive symptoms (20–24). Scores beginning at the moderate severity are considered clinically significant and warrant further evaluation for depression (α = .90). The PHQ-8 is widely used and has demonstrated evidence to support reliability and validity, as well as clinical utility of cut-off scores [16].

**Impact of events scale- revised.** The Impact of Events Scale- Revised [IES-R; 17] is a 22-item measure assessing post-traumatic stress symptoms with respect to a specific traumatic

**Table 1. Sample demographic characteristics.**

| Characteristic | Mean (*SD*) | N (%) |
|---|---|---|
| Relationship to child (% mothers) | | 89 (86.4%) |
| Parent age (years) | 38.39 (8.70) | |
| Parent Ethnicity | | |
| Non-Hispanic | | 24 (32.9%) |
| Hispanic | | 49 (67.1%) |
| Parent Race | | |
| White | | 59 (80.8%) |
| Black/African American | | 10 (13.7%) |
| Asian | | 1 (1.4%) |
| Other | | 3 (4.1%) |
| Parent marital status | | |
| Single | | 23 (22.3%) |
| Living with partner | | 7 (6.8%) |
| Married | | 64 (62.1%) |
| Separated | | 4 (3.9%) |
| Divorced | | 4 (3.9%) |
| Widowed | | 1 (1.0%) |
| Parent highest education obtained | | |
| Less than high school | | 2 (1.9%) |
| High school | | 3 (2.9%) |
| High school graduate | | 8 (7.8%) |
| GED | | 3 (2.9%) |
| Technical School | | 6 (5.8%) |
| Some college- no degree | | 17 (16.5%) |
| Associate degree | | 16 (15.5%) |
| Bachelor's degree | | 30 (29.1%) |
| Graduate degree | | 18 (17.5%) |
| Parent current employment status | | |
| Working on-site | | 23 (22.3%) |
| Working from home | | 29 (28.2%) |
| Combination of on-site and from home | | 15 (14.6%) |
| Furloughed | | 5 (4.9%) |
| Laid off | | 6 (5.8%) |
| Unemployed (pre-pandemic) | | 19 (18.4%) |
| Retired | | 2 (1.9%) |
| Other | | 4 (3.9%) |
| Number of children living in the home | 1.96 (.95) | |
| Child age (years) | 9.72 (4.16) | |
| Child device type | | |
| Cochlear implant | | 54 (52.4%) |
| Hearing aid | | 39 (37.9%) |
| Bone anchored device | | 7 (6.8%) |
| None | | 3 (2.9%) |
| Survey language (% English) | | 85 (82.5%) |
| COVID-19 Exposure | | |
| Someone in the family. . . | | |
| Was exposed to someone with COVID | | 10 (9.9%) |

(*Continued*)

**Table 1.** (Continued)

| Characteristic | Mean (*SD*) | N (%) |
|---|---|---|
| Had symptoms or was diagnosed | | 9 (8.9%) |
| Was hospitalized | | 4 (4%) |
| Was in the ICU | | 2 (2%) |
| Died from COVID-19 | | 1 (1%) |

event. For the purposes of our study, we included COVID-19/the pandemic as the traumatic event to base answers on. Items are rated on a 5-point Likert scale (0 = not at all to 4 = Extremely). Higher scores are indicative of increased post-traumatic stress symptoms (α = .96). Scores 24 and above indicate that Post-traumatic Stress Disorder is a clinical concern.

## Statistical analyses

Analyses were conducted using the Statistical Package for Social Sciences, version 26 [18]. There was limited missing data and Little's MCAR test revealed that data were missing completely at random ($\chi2 = 5.96$, p = .43). Preliminary analyses were conducted between demographic variables and all outcome variables to identify any associations. No significant associations in parental mental health were found based on relationship to child, age, highest education, or marital status. Next, descriptive analyses were conducted for demographic and outcome variables. Regression analyses were then conducted between family exposure and impact of COVID-19, COVID-19 related distress, and mental health symptoms (e.g., anxiety, depression, and PTSD).

## Results

Overall, the sample mean for the GAD-7 was within the moderate range (*M* = 5.55). However, when examining the sample by severity, 41% endorsed symptoms in the moderate range, 7% in the moderate to severe range, and 7% in the severe range. In contrast parents reported less symptoms of depression, with the sample mean in the mild range (*M* = 5.82). Specifically, 8% endorsed symptoms in the moderate range, 5% in the moderate to severe range, and 3% in the severe range. In terms of PTSD symptoms, the overall sample was below the clinical cut-off

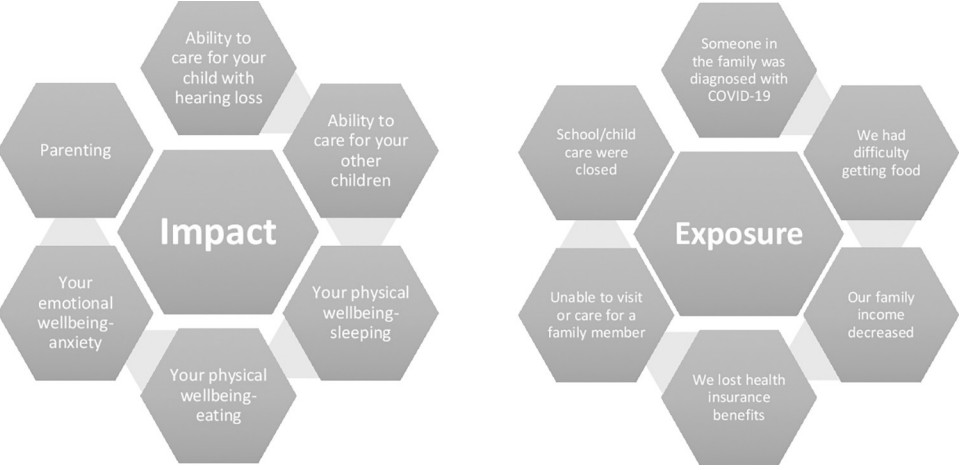

**Fig 1. COVID-19 Exposure and Family Impact Survey (CEFIS).** The Impact and Exposure scales of the CEFIS.

($M$ = 16.31). However, evaluation of clinical concern revealed that 20% of the participants endorsed PTSD symptoms above the clinical concern range on the IES, with 10% scoring in the highest range indicating that symptoms are high enough to suppress immune functioning and warrant further evaluation and intervention. No differences in parent report of mental health symptoms were observed based on parent employment status, children's degree of hearing loss or device type. While no statistical differences were able to be analyzed due to the differences in group size by race, a trend was observed for African-American/Black individuals reporting a higher mean for PTSD (African-American/Black $M$ = 27.0, White $M$ = 14.7). No statistical differences in depression, anxiety, or PTSD were observed by ethnicity.

## COVID-19 models: Effects of impact versus exposure

Linear regressions were conducted to examine COVID-19 impact and exposure and parent mental health (e.g., anxiety, depression, and PTSD symptoms). Anxiety levels were significantly related to COVID-19 impact [$F(2,98)$ = 8.92, $p < .01$], with parents who reported higher impact ratings endorsing more anxiety symptoms ($B$ = .24, $p < .001$). Anxiety levels were not related to COVID exposure. In contrast, depression levels were significantly associated with COVID-19 impact and exposure [$F(2,98)$ = 22.71, $p < .01$], with parents who reported more impact ($B$ = .40, $p < .001$) and more exposure ($B$ = .37, $p < .05$) endorsing elevated depressive symptoms. See Table 2 for descriptives of outcome variables. Similarly, PTSD symptoms were significantly associated with COVID-19 impact and exposure [$F(2,97)$ = 14.16, $p < .01$], with parents who reported more impact ($B$ = .94, $p < .001$) and more exposure ($B$ = 1.42, $p < .05$) endorsing increased PTSD symptoms. See Table 3 for models predicting mental health symptoms.

Furthermore, linear ***regressions*** were conducted to examine the associations between family exposure and impact of COVID-19 on COVID related distress for parents and children, respectively. Impact and exposure [$F(2,100)$ = 11.96, $p < .01$], were associated with parent COVID related distress, with parents who reported more impact ($B$ = .12, $p < .001$) and more exposure ($B$ = .14, $p = .058$, marginal) endorsing higher COVID related distress. However, for children only impact had a significant association with COVID related distress [$F(2,99)$ = 7.87, $p < .01$], with parents who reported more impact endorsing higher COVID related distress in their children ($B$ = .12, $p < .01$).

## Discussion

The COVID-19 crisis has disproportionately affected minority families and vulnerable populations, as communities of color or lower socioeconomic backgrounds have higher rates of

**Table 2. Descriptives of outcome variables.**

| Outcome | Mean (*SD*) | Sample Min-Max |
|---|---|---|
| GAD-7 | 5.55 (8.70) | 0–21 |
| PHQ-8 | 5.82 (8.70) | 0–24 |
| IES | 16.31 (8.70) | 0–88 |
| CEFIS- impact | 22.61 (6.49) | 10–37 |
| CEFIS- exposure | 7.99 (8.70) | 2–14 |

*Note*. GAD clinical cutoff: (0–4) mild, (5–9) moderate, (10–14) moderately severe, and (15–21) severe symptoms; PHQ-8 clinical cutoff: (5–9) mild, (10–14) moderate, (15–19) moderately severe, and (20–24) severe symptoms; IES clinical cutoff: (24–32) PTSD is a clinical concern, (33–38) probable diagnosis of PTSD, and (39+) elevated enough to suppress immune system's functioning.

**Table 3. Models predicting mental health symptoms.**

|  | β | B (SE b) | 95% CI | Model R2 | F |
|---|---|---|---|---|---|
| *Model 1: Anxiety* |  |  |  |  |  |
| CEFIS- impact | .33** | .24 (.07) | .10, .38 | .16 | 8.92** |
| CEFIS- exposure | .15 | .26 (.17) | -.09, .60 |  |  |
| *Model 2: Depression* |  |  |  |  |  |
| CEFIS- impact | .49** | .40 (.07) | .26, .54 | .32 | 22.71** |
| CEFIS- exposure | .19* | .37 (.17) | .03, .72 |  |  |
| *Model 3: PTSD* |  |  |  |  |  |
| CEFIS- impact | .37** | .94 (.24) | .46, 1.41 | .23 | 14.16** |
| CEFIS- exposure | .23* | 1.42 (.58) | .26, 2.57 |  |  |

Note $**p < .001$, $*p < .05$.

infection and mortality [19–21]. However, little attention has been placed on families at increased risk for parenting distress or mental health concerns, such as families of children with chronic conditions or developmental disorders. The current study utilized a sample of parents of children with hearing loss to identify the key factors predicting mental health concerns resulting from the COVID-19 pandemic.

In our study of 103 parents of children with varying degrees of hearing loss, we found that 55% of parents reported clinically elevated symptoms of anxiety during the pandemic, with approximately 7% in the severe range. According to the Anxiety & Depression Association of American, this is more than three times higher than the estimated prevalence of 19% of clinically diagnosed anxiety in the general population [22]. Furthermore, 16% of parents endorsed elevated symptoms of depression, including feelings of hopelessness, sadness, and difficulties sleeping and eating. In comparison to the prevalence of depression (3.8%) in the general population, the parental report in the current study was nearly four times higher than current estimates for clinical depression [23]. While our rates are higher than estimates in the general population, they are similar with studies reporting on changes in depression from pre- to during the pandemic. A recent study using the same measure utilized in the current study, the Patient Health Questionnaire, reported a three-fold increase in depression from pre- to during COVID-19 [8.5% to 27.8%; 24]. While prior studies have noted more mental health concerns for parents of younger children, our study found that the age of the child did not affect mental health symptomatology for our parents of children with hearing loss. This suggests parents of children of any age are at risk for mental health concerns while living through a global pandemic. Generally, a decline in parental mental health is strongly related to child behavioral health, with four in every 10 adults reporting worsening of mental health during the pandemic [25].

Specifically in the hearing loss population, it has been reported that children of depressed parents use their hearing devices approximately two and a half hours per day less than children with parents who reported no depression [26,27]. Moreover, history of parent depression has been shown to be the strongest predictor of missed medical appointments for children [28]. Thus, identification and management of mental health symptoms in parents of children with hearing loss is drastically warranted as reduced wear time of hearing devices and consistent absences from audiology and speech appointments is detrimental to the child's overall success with hearing technology [29–31].

According to the CDC those with multiple stressors or persons with chronic medical illnesses are at greater risk for developing long-term reactions, such as PTSD [32]. PTSD is

characterized by intense physical and emotional reactions to thoughts or reminders of an event, including avoidance, re-experiencing and arousal symptoms [33]. If left untreated, long-term effects of PTSD can include an increase in activity of the sympathetic nervous system and decrease of the parasympathetic nervous system, resulting in substantial distress and disruption of social and occupational functioning. In our study, parents of children with hearing loss are among those who are at increased risk for PTSD as 20% of parents reported symptoms consistent with PTSD, with 10% reporting severe symptoms warranting a diagnosis. According to the National Institute of Mental Health, 3.6% of individuals will have a PTSD diagnosis in a given year, with a lifetime prevalence of 6.8% [34]. Thus, our sample of 103 parents of children with hearing loss endorsed symptoms consistent with PTSD that were three to five times higher than expected compared to the general population. To date, this is the first study to evaluate PTSD symptomatology during the pandemic in parents of children who are considered vulnerable. A recent study focusing on parents in the general population reported that 25% of parents met criteria for PTSD during quarantine and 86% of those parents had children who also met the clinical cut-off scores [35]. However, our findings are aligned with more recent studies documenting that 30% of individuals are endorsing symptoms consistent with PTSD after acute COVID-19 infection [36].

While previous studies have reported that minority groups were disproportionately affected by the pandemic, our study did not find any overall differences by race or ethnicity in terms of impact or exposure to COVID-19. However, in our small sample of African-American/Black parents ($n$ = 10) a higher mean for PTSD was found compared to White parents. This is interesting as no differences were found in impact or exposure of COVID-19 by race suggesting that while African-American/Black families may have had similar exposure and impact, as well as similar elevations in mental health (anxiety, depression) symptoms they perceived the event as more traumatic compared to other minority groups. Hispanic and African-American/Black individuals are generally less affected by mental health disorders, however, when they do have a mental health condition they have worse symptoms and outcomes and are less likely to seek treatment [37]. Thus, this highlights the need for continuous mental health screening across all populations as all racial groups appear affected by the global pandemic. The presence of hardships has also been consistently linked with worse mental health [2,38,39]. The current study both confirmed the effects of the pandemic on mental health, as well as identified the key factors and exact hardships that contribute to parental mental health difficulties. Interestingly, anxiety was only predicted by impact, meaning that an individuals' perception of their ability to care for their children, parenting, as well as their sleeping, eating and overall wellbeing affected their report of anxiety symptomatology. Alternatively, depression and PTSD were associated with both impact and exposure to COVID-19. Thus, depression and PTSD were affected by an individuals' actual exposure to someone in their family with COVID, inability to visit family members, or school closures, as well as their perceptions of their overall wellbeing (sleep, eating, emotional adjustment), and ability to care for their children. These results are consistent with a large study of US adults reporting that greater exposure to COVID-19 stressors predicted higher rates of depression [24].

Additionally, overall parent distress was influenced by both exposure and impact. Thus, it appears that for our specialized population of parents of children with hearing loss, the impact of the pandemic may have had a greater effect on mental health than exposure alone. Our data suggest that infection, community shutdowns, and income loss, were not solely responsible for overall mental health difficulties. Rather, COVID-19 impact in terms of parent ability to care for their child, as well as sleeping, eating, and general parenting skills, were the more prominent factors in this study influencing anxiety, depression, or PTSD symptomatology. Thus, the effects of the pandemic are not only the quantity of hardships that families experience, but the

extent to which those hardships impacted parenting in our sample of parents of children with hearing loss. Findings are consistent with prior research reporting that COVID-19 did not predict maternal distress, rather distress is more affected by how the exposure impacted your life, as well as coping [40].

Despite these higher rates of mental health concerns, past research has shown the power of resilience and how families experiencing hardships may use these experiences as an opportunity to grow and improve their parent-child relationship [41–43]. For instance, Hurricane Sandy was associated with increased rates of depression and stress; however, data also identified the importance of community and individual resources to build post-disaster resilience [42]. Future research efforts should focus on post-pandemic challenges, including long-term psychological effects, such as depressive disorders and PTSD. Although health and economic hardships may continue, exploration of the long-term psychological sequelae on parent mental health, the extent to which these symptoms remain post-pandemic, and how they may influence long-term child mental health and development is still needed.

Given the known influence that parents have on their children's well-being, academic and medical professionals (pediatricians, primary care physicians, and mental health providers) need to work collaboratively to establish guidelines on how to appropriately identify parents who are experiencing clinically significant anxiety, depression, or PTSD. This is particularly important as parents of children with chronic medical conditions and/or developmental disorders are critical members of the treatment team that have been shown to influence long-term treatment outcomes [44–46]. Further investigation of the effects of parent mental health on child outcomes is necessary in order to influence our clinical recommendations in the hearing loss population. With the increase in telemedicine, institutions and community centers should continue to offer multiple service delivery options, such as remote (i.e., phone, online) or direct intervention for psychological/psychiatric support for parents of children with hearing loss as they are at higher risk for mental health concerns.

## Limitations

Although this is the first study to evaluate the impact and exposure of COVID-19 on parent mental health (e.g., depression, anxiety, and post-traumatic stress), for parents of children with hearing loss, there are also important limitations to consider. First, the CEFIS was not a validated tool at the time of administration given that it was created at the beginning of the pandemic. However, since then, it has been psychometrically validated with deidentified data from over 28 programs at 15 institutions across the United States. Additionally, the CEFIS is accessible via the National Institutes of Health Disaster Information Management Research Center. Another limitation is that the survey was only open for a one-month period. Given that this was at the beginning of the pandemic when many things were unknown, the authors closed the survey in order to prevent having it open near the end of the pandemic or the original quarantine periods that were expected to be approximately two weeks. Furthermore, having a tight window allowed for the responses of parents to be more cohesive and representative of the pandemic timepoint. If the authors would have left the survey open for several months during this period of uncertainty and constant changing environment, the data may have differed for those at the beginning versus those at the end of the survey window. Moreover, 69% of parents in this study reported that their mental health was made worse due to the pandemic and these findings are consistent with larger national studies evaluating the effects of the pandemic on mental health. Lastly, due to the research limitations during the initial stages of the pandemic, an electronic questionnaire was utilized, thus limiting our collection of specific audiological information (i.e., audiometric thresholds, device use, duration of hearing loss),

language outcomes, treatment/appointment adherence, or rehabilitative services. Thus, preventing the evaluation of the potential mediational effects of audiological information and/or intervention on parental wellbeing. Future research should evaluate the impact of the pandemic on audiological and educational outcomes, as well as the inherent effects of children's rehabilitation on parent mental health.

In conclusion, this study highlighted the mental health impact of COVID-19 on parents of children with hearing loss and underscored the importance of including mental health screening and treatment into hearing healthcare. Due to the established relationship between parental mental health and pediatric hearing outcomes, providers should ensure that family support services, including psychology and social work, are available for parents and children who need them.

## Supporting information

**S1 File.**
(XLSX)

## Author Contributions

**Conceptualization:** Ivette Cejas, Jennifer Coto.

**Investigation:** Ivette Cejas.

**Methodology:** Ivette Cejas, Jennifer Coto.

**Writing – original draft:** Ivette Cejas, Chrisanda Sanchez, Meredith Holcomb, Jennifer Coto.

**Writing – review & editing:** Ivette Cejas, Chrisanda Sanchez, Meredith Holcomb, Jennifer Coto.

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
