## [Decision Letter · Decision Letter 0]

19 Sep 2022

PONE-D-22-12267Parents of children with hearing loss: Impact and exposure of COVID-19 on psychological well- beingPLOS ONE

Dear Dr. Cejas,

Thank you for submitting your manuscript to PLOS ONE. I have received two reviews from experts in the field, as well as carefully considered the manuscript myself. We all agree that your manuscript addresses an important topic; although, there are several areas of concern that will need to be addressed prior to publication. Both reviewers noted the need for greater clarity throughout the manuscript. This includes increased clarity in some of the terms used and their underlying constructs, but also how they are being operationalized and factored into your regression models. Both reviewers also noted some methodological concerns that will need to be addressed. While I understand that some methodological decisions may not be altered at this point, the limitations that they introduce should be acknowledged in the manuscript. Related to this, presentation of the findings should accurately align with your results. Finally, please edit the manuscript for clarity throughout. Therefore, we invite you to submit a revised version of the manuscript that addresses the points raised during the review process.

We look forward to receiving your revised manuscript.

Kind regards,

Eric J. Moody, Ph.D.

Academic Editor

PLOS ONE

a) Did participants provide their written or verbal informed consent to participate in this study?

Reviewers' comments:

Reviewer's Responses to Questions

**Comments to the Author**

1. Is the manuscript technically sound, and do the data support the conclusions?

Reviewer #1: Partly

Reviewer #2: Partly

2. Has the statistical analysis been performed appropriately and rigorously? 

Reviewer #1: No

Reviewer #2: Yes

3. Have the authors made all data underlying the findings in their manuscript fully available?

Reviewer #1: Yes

Reviewer #2: Yes

4. Is the manuscript presented in an intelligible fashion and written in standard English?

Reviewer #1: Yes

Reviewer #2: Yes

5. Review Comments to the Author

Reviewer #1: Dear Authors

This work focuses on a topic that is certainly very interesting and topical, and which highlights the need to deepen the mental health of parents of children with hearing loss, especially in these difficult times we are experiencing due to the pandemic.

The paper is very well written and overall clear. However, there are some aspects that need to be deepened and integrated to make the study more relevant.

In terms of content, it would be interesting if some aspects were more investigated. For example, the introduction is a bit repetitive and not very in-depth.

As far as the survey is concerned, some items could be reported as examples (perhaps the most significant ones or those with the most relevant scores).

Within the limits of the work, it is pointed out that it was not possible to retrieve data on children's audiometric thresholds. In addition to these data, however, it would have been important to have information regarding the etiology of hearing loss, any associated disabilities, the progress of rehabilitation (perceptual and language outcomes), perhaps even to know how long the child has been wearing the device, whether it is a prosthetic or cochlear implant. The "child device type" alone is likely to be a variable that does not allow for in-depth analysis.

Equally interesting would be to have the same data for a sample of parents of children without hearing impairment in order to see how much the children's condition affects the parent's status at such a critical time.

When in the discussion of making a comparison with similar assessments of parents of normal hearing children (e.g., "This is more than three times higher than the estimated prevalence of 18% in the general population"), what tools were used in the cited work? Does this refer to a clinical diagnosis of depression?

Lines 221-224 should be more clearly analyzed on the one hand the relationship between general population and parents involved in the study and, on the other hand, between the pre- and post-covid situation, and then discuss the data together.

Lines 282-303 better argue the discussion; at some points, it does not appear to be sufficiently supported by the results.

Why did only one parent per child participate in the study? It would also be interesting to see if there are differences between the different parental figures.

It would be better to define the acronym Post-traumatic stress disorder after the first time the acronym is reported.

Reviewer #2: The reported aim of this study " to evaluate the impact and exposure of COVID-19 on parent mental health (e.g., depression, anxiety, and post-traumatic stress), for parents of children with hearing loss" is a timely and important topic. The authors do a nice job of positioning this paper within the context of covid-19 and in describing results situate the findings nicely by drawing similarities with prior research (e.g., findings re: trends in race for African American/black parents and impact of covid; rates of depression in this group compared with general global findings).

A major concern with this paper is that the aims and how the IV and DVs are measured is not always clear. At various points the authors refer to the construct of "well being" but this is not clearly defined, so it is unclear whether this is referring to "mental health" or to the construct of well being as measured by items from the CEFIS. Your aims are to measure “impact and exposure” of COVID 19 (CEFIS scales) on parent mental health (GAD7, PHQ, IESR) so it is very confusing to conflate mental health measures with “well being” when CEFIS items directly assess questions related to well being (which is measured by their part 2- impact scale). Similarly, in the literature review, papers referring to parental depression are instead described as assessing "well being." This needs to be clarified and corrected throughout the paper.

A striking example of this confusion of terms comes from Lines 74-77:

" This study evaluates the impact (WELLBEING, emotional adjustment, and family relationships) and exposure 75 (school closures, changes in housing, changes in employment) of COVID-19 on the overall 76 WELLBEING of parents of children with hearing loss, as defined by symptoms of depression, 77 anxiety, and post-traumatic stress disorder.

This would suggest that you are measuring wellbeing as both an IV and DV?

Another limitation of this study is that the CEFIS impact scales have items measuring anxiety. Therefore, findings regarding linear regression that were found to be significant between the GAD7 and CEFIS impact scales may reflect overlapping constructs rather than overall "impact" predicting anxiety per se.

Furthermore, it really needs to be clarified whether you are using the CEFIS to measure 3 independent variables (exposure and impact) or 3 (exposure, impact, and distress). If I’m reading this correctly, after looking closely at the CEFIS measure, it seems like you are measuring parent distress and child distress based on a SINGLE item from the CEFIS? This does not seem like one of your initial aims and seems methodologically flawed.

From line 34: "In addition, both impact and exposure predicted COVID related parental distress, while child distress was only affected by impact of COVID-19"

The following lines seem to conflate impact and distress in a way that is confusing:

Lines 118 -120…." open-ended question inquiring about the family’s experience and effects of COVID. Higher scores on both the exposure and impact scales denote more negative exposure and impact/distress. (Are these the same construct? or different constructs?). In general, I find the interpretation of data related to child distress to be weak given the limited data available to adequately assess child distress and since this is not a primary aim of the study, I would recommend dropping this discussion from the paper.

A methodological limitation of the study not well discussed in the discussion is the availability of the survey for a very short window (one month) at the beginning of COVID19 (June 9th, 2020- July 9th 91 , 2020) at the height of uncertainty related to COVID. There is little discussion about why this window was chosen. It would have been of more practical value to reassess respondents at a later timepoint in order to better ascertain the persistence of these elevated symptoms. Failure to do this means caution must be used in drawing conclusions from this data particularly when extrapolating beyond the data regarding suggested implications for intervention (e.g., lines 309-310 regarding necessity for telehealth). Discussion sections such as this should be edited to more clearly delineate what can be clearly drawn from the actual data presented, versus possible implications.

Some of the interpretation of the data is somewhat misleading. For example, the authors cite that " 20% of the participants endorsed PTSD symptoms above the clinical concern 168 range on the IES, with 10% scoring in the highest range indicating that symptoms are high 169 enough to suppress immune functioning and warrant further evaluation and intervention." This is not adequately captured in the TABLE 2 where the mean score is reported as falling below the clinically significant range for PTSD (WHICH IS NOT DISCUSSED) but there is quite a range. This seems to misrepresent or not adequately capture the data in its entirety. What you’re finding here is that the average is below the clinical cut off, and with this small sample size that 20% falling above the clinically significant range is harder to interpret. It’s also unclear to me whether the Min-Max represents min-max possible scores or the range of scores for this sample- please consider relabeling.

The literature review and discussion sections should be edited for clarity. The authors attempt to weave together literature that children who are DHH are more at risk for mental health concerns, that parents of DHH children are more vulnerable to parenting stress and depression impacting caregiver sensitivity and language outcomes, and that children of depressed parents are more vulnerable to poor treatment adherence. However, both the literature review and discussion sections, appear disjointed. This is especially true in the discussion section where the authors link findings that in the general population parental depression is correlated with missed appointments and findings that poor hearing device compliance is linked with poor outcomes. (231-236) However, a transitive property is being utilized here that was not directly measured in this study. (If data regarding treatment adherence during this time period was available, it would strengthen the paper). As presented, this feels like an overextension and should be presented as a possible concern to be explored in future research (e.g., it's unclear whether elevated depressive symptoms during a pandemic, especially when it is unclear how persistent those symptoms are, will have same implications for treatment adherence as past research on parents with depression meeting clinical criteria for depression).

Finally, there are several places where awkward syntax is used. Editing language would be helpful. Specific examples (not exhaustive) include:

Line 4 – awkward structure requires revision

Line 71-73"As the world prepares for post-pandemic, it is crucial that we begin to identify and understand how mental health has been affected families of children with hearing loss in order to influence our clinical recommendations and guide our medical treatment decisions.”

Line 246 -249 ..."system. Further, PTSD can get worse if not addressed, resulting in substantial distress and disruption of social and occupational functioning. According to the CDC those with multiple stressors or persons with chronic medical illnesses are at greater risk for developing long-term reactions, such as PTSD [37]" ( latter sentence here may be better placed before the first?)

6. PLOS authors have the option to publish the peer review history of their article (what does this mean?). If published, this will include your full peer review and any attached files.

Reviewer #1: No

Reviewer #2: No

---

## [Author Response · Author response to Decision Letter 0]

1 Dec 2022

All the response to reviewers has been included as an uploaded document.

---

## [Decision Letter · Decision Letter 1]

14 Feb 2023

PONE-D-22-12267R1Parents of children with hearing loss: Impact and exposure of COVID-19 on mental healthPLOS ONE

Dear Dr. Cejas,

Thank you for submitting your manuscript to PLOS ONE. The previous reviewers and I have reviewed your revised manuscript, and all of us agree it is significantly improved. However, you will see that the reviewers have a few remaining concerns, some of which are merely editorial.  I suspect that the requested revisions will be relatively easy to respond to, and I look forward to receiving your updated manuscript at your earliest convenience. 

We look forward to receiving your revised manuscript.

Kind regards,

Eric J. Moody, Ph.D.

Academic Editor

PLOS ONE

Journal Requirements:

Reviewers' comments:

Reviewer's Responses to Questions

**Comments to the Author**

1. If the authors have adequately addressed your comments raised in a previous round of review and you feel that this manuscript is now acceptable for publication, you may indicate that here to bypass the “Comments to the Author” section, enter your conflict of interest statement in the “Confidential to Editor” section, and submit your "Accept" recommendation.

Reviewer #1: All comments have been addressed

Reviewer #2: All comments have been addressed

2. Is the manuscript technically sound, and do the data support the conclusions?

Reviewer #1: Partly

Reviewer #2: Partly

3. Has the statistical analysis been performed appropriately and rigorously? 

Reviewer #1: Yes

Reviewer #2: Yes

4. Have the authors made all data underlying the findings in their manuscript fully available?

Reviewer #1: Yes

Reviewer #2: Yes

5. Is the manuscript presented in an intelligible fashion and written in standard English?

Reviewer #1: Yes

Reviewer #2: Yes

6. Review Comments to the Author

Reviewer #1: Dear authors,

I renew my congratulations for the research done and the interest in the topic.

In the case of children with hearing loss, we often underestimate the fundamental importance that the emotional and mental health status of the parents has on the child's outcomes and rehabilitation, as well as the child's approach toward the use of devices such as prostheses or cochlear implants.

And a situation like the covid-19 pandemic may necessarily have altered all of these factors.

I appreciated the authors' effort to respond to all the reviewers' comments. What concerns the actual evaluation of the parameters reviewed is now generally clearer to me, thanks to the correction of terminology that was previously somewhat confusing.

So, too, the (perhaps not entirely agreeable) choice of the selected one-month time window is better argued.

Overall, this remains a work that in terms of discussion proceeds more by hypothesis than by evidence. Therefore, it is fine to have "lightened" the discussion of the results, no longer giving interpretations that were not fully supported by the data. However, many perspectives remain open for further investigation (necessary, as far as I am concerned, in order to better analyse the issue under discussion).

In any case, the authors have better pointed out these gaps and highlighted the study's weaknesses, to be resolved perhaps by further studies.

Regarding especially the absence of correlations between questionnaire scores and children's audiological variables, it is obvious how these may be impacted by the limitations imposed by the pandemic. But my comment in the previous review was not so much about the influence of the pandemic on these variables, but the opposite, namely the possible mediating role that these variables may have on parental mental health. In fact, I imagine that the child's rehabilitation (audiological outcome, level of compensation of hearing loss, time of device use, results of language rehabilitation, child's listening ability, frequency of clinical follow-ups...) are closely related to the consequences that all covid-19 limitations had on the well-being of the parents of these children, and are therefore inescapable variables in the perspective of a more correct analysis of the results.

Therefore, I would better emphasise this among the limitations and future prospects of the study.

Overall, as the first study to analyse this topic, the article is now clearer and more coherent and of potential interest to those dealing with hearing impaired patients and their families.

Reviewer #2: The author revisions have greatly improved the article and particularly ensured that conclusions drawn are better aligned with findings. A few points still worth clarifying/editing:

Lines 75-77 : This study evaluated the impact (MENTAL HEALTH, emotional adjustment, and family relationships) and exposure (school closures, changes in housing, changes in employment) of COVID-19 on the overall mental health of parents of children with hearing loss, as defined symptoms of depression, anxiety, and Post-Traumatic Stress Disorder (PTSD).

- These lines should be rewritten. As written, it sounds like you are looking at relationships between Impact (MENTAL HEALTH)…. “on the overall MENTAL HEALTH” of parents of children with HL. In the text beginning on line 152 impact is summarized as “individuals’ overall wellbeing, emotional adjustment, sleeping, and ability to care for their children”- please edit for consistency

- Also lines 158-161 add further confusion by adding in “distress” which is not mentioned earlier and seems to introduce a 3rd construct

Technical edits:

Line 142 – spell out acronym first (not just in the header) “The CEFIS”

Line 342 “Thus, it appears that for our specialized population of parents of children with hearing, …” (insert …children with (reduced?) hearing??? Or “children with hearing (loss?))

7. PLOS authors have the option to publish the peer review history of their article (what does this mean?). If published, this will include your full peer review and any attached files.

Reviewer #1: No

Reviewer #2: No

---

## [Author Response · Author response to Decision Letter 1]

7 Mar 2023

Response to the reviewers have been uploaded as a separate document. All Reviewer comments were addressed and have been highlighted in the text by track changes.

---

## [Decision Letter · Decision Letter 2]

19 Apr 2023

Parents of children with hearing loss: Impact and exposure of COVID-19 on mental health

PONE-D-22-12267R2

Dear Dr. Cejas,

We’re pleased to inform you that your manuscript has been judged scientifically suitable for publication and will be formally accepted for publication once it meets all outstanding technical requirements.

Kind regards,

Eric J. Moody, Ph.D.

Academic Editor

PLOS ONE

Additional Editor Comments (optional):

Reviewers' comments:

Reviewer's Responses to Questions

**Comments to the Author**

1. If the authors have adequately addressed your comments raised in a previous round of review and you feel that this manuscript is now acceptable for publication, you may indicate that here to bypass the “Comments to the Author” section, enter your conflict of interest statement in the “Confidential to Editor” section, and submit your "Accept" recommendation.

Reviewer #1: All comments have been addressed

2. Is the manuscript technically sound, and do the data support the conclusions?

Reviewer #1: Partly

3. Has the statistical analysis been performed appropriately and rigorously? 

Reviewer #1: Yes

4. Have the authors made all data underlying the findings in their manuscript fully available?

Reviewer #1: Yes

5. Is the manuscript presented in an intelligible fashion and written in standard English?

Reviewer #1: Yes

6. Review Comments to the Author

Reviewer #1: The manuscript, already greatly improved after previous revisions, is now complete, and the last few small details have also been corrected. The topic analyzed, as previously written, is very interesting and opens up prospects for further study. The weaknesses of the study, from my point of view, are not resolvable as they would need further study, new data collection, and new analysis, but I appreciate that they have now been more explicitly described in the section on the limitations of the work.

7. PLOS authors have the option to publish the peer review history of their article (what does this mean?). If published, this will include your full peer review and any attached files.

Reviewer #1: No
